# The AI-Assisted Identification and Clinical Efficacy of Baricitinib in the Treatment of COVID-19

**DOI:** 10.3390/vaccines10060951

**Published:** 2022-06-15

**Authors:** Peter J. Richardson, Bruce W. S. Robinson, Daniel P. Smith, Justin Stebbing

**Affiliations:** 1Benevolent AI, 4-8 Maple Street, London W1T 5HD, UK; dan.smith@benevolent.ai; 2UWA School of Medicine, Perkins Institute for Medical Research, Sir Charles Gairdner Hospital, Nedlands, Perth 6009, Australia; bruce.robinson@uwa.edu.au; 3Imperial College, Hammersmith Hospital, London W12 0NN, UK; j.stebbing@imperial.ac.uk

**Keywords:** COVID-19, SARS-CoV-2, baricitinib, antiviral, anti-cytokine, immune modulator

## Abstract

During the current pandemic, the vast majority of COVID-19 patients experienced mild symptoms, but some had a potentially fatal aberrant hyperinflammatory immune reaction characterized by high levels of IL-6 and other cytokines. Modulation of this immune reaction has proven to be the only method of reducing mortality in severe and critical COVID-19. The anti-inflammatory drug baricitinib (Olumiant) has recently been strongly recommended by the WHO for use in COVID-19 patients because it reduces the risk of progressive disease and death. It is a Janus Kinase (JAK) 1/2 inhibitor approved for rheumatoid arthritis which was suggested in early 2020 as a treatment for COVID-19. In this review the AI-assisted identification of baricitinib, its antiviral and anti-inflammatory properties, and efficacy in clinical trials are discussed and compared with those of other immune modulators including glucocorticoids, IL-6 and IL-1 receptor blockers and other JAK inhibitors. Baricitinib inhibits both virus infection and cytokine signalling and is not only important for COVID-19 management but is “non-immunological”, and so should remain effective if new SARS-CoV-2 variants escape immune control. The repurposing of baricitinib is an example of how advanced artificial intelligence (AI) can quickly identify new drug candidates that have clinical benefit in previously unsuspected therapeutic areas.

## 1. Introduction

The COVID-19 pandemic has caused the death of approximately 6 million people, with a case fatality rate which may be as high as 20% in those over 80 years old [1]. Vaccines have proved to be extremely effective in reducing the damage and hospitalisation caused by this infection, although some patients still need supportive care. As the SARS-CoV-2 virus has continued to evolve, the potential for the virus to escape vaccine and exposure induced immunity remains a threat. In this situation, as at the start of the pandemic when no such vaccines were available, it is important that there exist therapeutics for the treatment of severely ill patients. This review described the identification, mechanism of action, and validation of the already approved rheumatology drug baricitinib as a treatment for hospitalised patients with COVID-19. In addition, comparison with other agents demonstrates that this drug is the most potent of the immune modulators in reducing COVID-19 mortality. As a result, it is now strongly recommended for the treatment of COVID-19 by the WHO.

Infection by SARS-CoV-2 is usually via respiratory droplets and, like the related SARS-CoV-1 and MERS viruses, results in a biphasic disease (Figure 1). The first phase shows mild symptoms, e.g., fever, muscle pains, fatigue, headache, diarrhoea, loss of taste and smell, and a cough which may last for up to 2 weeks. This is the experience of most patients, but in some this phase is followed by the onset of breathlessness and pneumonia, often requiring oxygen therapy, and which can also be associated with severe pulmonary and systemic inflammation, similar to a cytokine storm. This involves high levels of circulating cytokines with widespread organ damage, vascular damage/thrombosis, and acute respiratory distress syndrome (ARDS). It is unclear why some people suffer from this hyperinflammatory episode while others do not. Perhaps the most common explanation is that in those experiencing severe disease the response of both the innate and adaptive immune systems is dysregulated [2]. This dysregulated response is associated with ageing of the immune system, obesity [3], and with chronic underlying diseases such as cardiovascular disease, diabetes, COPD [4], and others.

The pathophysiological mechanisms implicated in COVID-19 include virus induced cytopathy, hypertension from virus induced internalisation of its receptor ACE2, hyperinflammation including cytokine and complement activation, cell death from excessive cytokines (pyroptosis), hypercoagulation, and perhaps autoimmunity [5]. The endothelial lining of the microvasculature appears particularly hard hit, probably due to a combination of factors including ACE2 expression with consequent virus infection, microthrombosis, and innate immune activation with neutrophil and macrophage activation and extravasation.

## 2. Host Susceptibility to SARS-CoV-2

### 2.1. Ageing and COVID-19

Ageing, the major risk factor for severe COVID-19, results in the accumulation of a number of defects in the innate and adaptive immune systems. For example, the number of T and B lymphocytes, macrophages, granulocytes, and lymphatic follicles are significantly decreased in the elderly. Ageing macrophages and granulocytes adopt an enhanced inflammatory state secreting pro-inflammatory cytokines, and showing impaired phagocytosis, migration, and clearance, thereby compromising the ability of these cells to clear infections and damage [6]. Thus, the aged cells of the innate immune system generate a proinflammatory state (so called ‘inflammaging’) associated with reduced clearance of virus and virus-infected cells. This may also be associated with the accumulation of senescent cells in other tissues where they secrete a range of mediators known as the senescence-associated secretory phenotype (or SASP [7]). These mediators include many proinflammatory cytokines which may also contribute to inflammaging.

Despite the low-grade inflammation seen in the aged, the development of excessive numbers of terminally differentiated T cells (particularly CD28^−^ CD27^−^ CD45RA^+^ CD8^+^ T cells), with a paucity of naïve T cells has been observed (a condition known as immunosenescence) [8]. The relative lack of naïve T cells compromises the ability of the aged immune system to mount a defence against novel pathogens such as SARS-CoV-2. Increased numbers of senescent T cells are also associated with autoimmune disease, chronic viral infection (e.g., CMV or EBV) [9], as well as the reduced response to vaccines seen in the aged [10,11]. These cells show increased NK receptor (e.g., KLRG-1), granzyme B, and perforin expression and have lost antigen-specific cell killing but retain a strong nonspecific killing potential [12].

Intriguingly, senescent T cells are also associated with some of the underlying medical conditions which increase the risk of severe COVID-19 disease. In rodent models, senescent T cells can induce diabetes and obesity, while their clearance moderates the disease [13,14]. It is therefore tempting to speculate that one reason for the susceptibility of the aged, and those with chronic diseases such as rheumatoid arthritis and diabetes, is due to the higher prevalence of senescent cells in these patients, including those of the immune system [15]. Similarly, T cell immunosenescence is closely related to the development of cardiovascular disease [16], another chronic disease state associated with susceptibility to severe COVID-19 disease [17]. However the role, if any, of senescent T cells in the susceptibility of the aged to SARS-CoV-2 has yet to be proven. 

### 2.2. Host Antiviral Responses to SARS-CoV-2

In the normal course of a viral infection, the innate immune system reacts first through the interferon (IFN) system, driven partly by recognition by cells of virus-related Pathogen Associated Molecular Patterns (PAMPs) which trigger a range of Toll Like (and other) Receptors (TLRs), [18], resulting in the expression of many IFN-stimulated genes (ISGs) and the stimulation of the adaptive immune response. The SARS-CoV-2 virus encodes a number of proteins, including ORF3b and ORF6, which inhibit this IFN response. Presumably this inhibition of the initial IFN response allows the virus to replicate in the early stages of the disease. Consistent with this, patients that have defects in the IFN system (e.g., anti-IFN autoantibodies) tend to experience severe disease and delayed viral clearance [19,20,21,22]. IFN response genetic variants have also been identified which are associated with severe disease [23], including the antiviral restriction enzyme activators OAS1, 2 and 3, the antiviral receptor TLR7 [24] and the IFN receptor IFNAR2. Further analysis suggested that high expression of the JAK enzyme Tyk2 (and the chemokine receptor CCR2) or low expression of IFNAR2 is associated with life critical disease [25]. Since the Type 1 IFN response is low in severe SARS-CoV-2 infections, but with elevated chemokines, it has been suggested that defective anti-viral responses result in excessive SARS-CoV-2 replication, elevated chemokines with consequent innate immune activation, and a resultant cytokine-mediated hyperinflammation [26]. In addition, the relative lack of naïve T cells in the elderly (and those with underlying health conditions such as autoimmune disease) may further reduce the ability of the adaptive immune system to respond to this new infection. This may then be amplified by the pro-inflammatory innate immune cells, and the senescent T cells which, while not expressing CD28 and CD27, have acquired NK cell properties and secrete yet more cytokines [27].

### 2.3. The Need for COVID-19 Therapeutics

Although the world has largely relied on vaccines for protection of our populations against SARS-CoV-2, there is still a need for therapies capable of reducing mortality in hospitalised patients, especially with the rapid evolution of new SARS-CoV-2 variants including the highly infectious Omicron variants. In addition, the resistance of a significant proportion of the population to being vaccinated, the escape of viruses from immune control, whether that be vaccine or infection induced, and the likelihood that other viruses with similar pathogenic mechanisms will be manifest in the future [28] means that there will be a need for COVID-19 therapeutics for some time to come.

Early in the pandemic there was an emphasis on testing already approved drugs in COVID-19 since this would be the fastest way of finding treatments which could be given to patients. The repurposing of anti-viral drugs such as hydroxychloroquine and Kaletra (lopinavir–ritonavir) was not a success with most agents not showing reproducible efficacy in clinical trials of hospitalised patients [29]. The anti-viral remdesivir was, however, approved in the USA for use as a result of the Adaptive Covid Treatment Trial-1 (ACTT-1), in which it reduced mortality and the duration of hospital stay [30]. It is now not recommended by the WHO for treating severely and critically ill patients but is for those at high risk of severe disease [31]. It has been difficult to prove that antiviral drugs are effective in COVID-19 due to the relatively mild symptomology during the incubation period, when the virus is most susceptible to such drugs. However, in many cases the viral load is already decreasing when patients come to hospital experiencing severe or critical disease, so immune modulators are required. This perhaps explains why the only drugs shown to reduce mortality in randomised clinical trials of hospitalised patients are anti-inflammatories [32]. The recent success of the antiviral Paxlovid was made possible by focusing on prospective patients at higher risk of hospitalisation due to underlying health conditions [33] and treating them in the early phase of the disease.

The first repurposed drugs to be approved for COVID-19 were the broadly acting immune-suppressive corticosteroids, which had modest beneficial effects on mortality and have been widely used throughout the world since mid-2020. The greatest effect of low dose dexamethasone on COVID-19 mortality was seen in the RECOVERY trial in patients receiving oxygen support or invasive mechanical ventilation. In patients with mild disease an increase in mortality was seen, perhaps partly as a consequence of suppression of the endogenous anti-viral immune response. This indicates that the timing of steroid (and other immune suppressant) administration may be critical [34]. There followed a very large number of clinical trials assessing various immune modulators on hospitalised patients with varying degrees of success. Amongst the agents tested were inhibitors of the JAK enzymes which mediate cytokine receptor signalling.

### 2.4. JAK Enzymes

The JAKs are tyrosine kinases activated by over 50 different cytokines via their receptors including IFNα, IFNγ, TNFα, IL-1β, IL-2, IL-6, and IL12. Activated JAKs phosphorylate intracellular surfaces of the receptors, promoting Signal Transducer and Activator of Transcription (STAT) binding and subsequent activation again through phosphorylation. The activated STAT proteins then regulate gene transcription in the nucleus. The four JAK enzymes (JAK1, JAK2, JAK3, and TYK2) activate seven STAT family members, the outcomes largely depending on the specific combination of JAKs and STATs activated by any given receptor [34]. In addition, a number of growth factors and related molecules use the JAK/STAT pathways including leptin, erythropoietin, thrombopoietin, and granulocyte–macrophage colony-stimulating factor (GM-CSF) [35].

JAK inhibitors including baricitinib, tofacitinib, peficitinib, upadacitinib, and filgotinib have been approved for the treatment of autoimmune diseases such as rheumatoid arthritis where they act as disease modifiers [35]. These approved inhibitors are all competitive with ATP showing varying amounts of selectivity between the JAKs based on in vitro assays. However, as discussed by Tanaka et al. [35], the in vitro selectivity does not always translate to predictable differences in cell-based cytokine signalling. JAK inhibitors such as ruxolitinib have also been developed for the treatment of myeloproliferative diseases which can be driven by mutations in JAK2 [36], and in which thrombosis is a major cause of mortality [37].

### 2.5. Virus Endocytosis

The entry of viruses into cells is mediated by a number of routes including Clathrin-Mediated Endocytosis (CME), caveolin-mediated endocytosis, macropinocytosis, and some other poorly described non clathrin- or caveolin-mediated mechanisms [38]. By far the best understood of these pathways is CME, which mediates the internalisation of many ligands with their receptors including the EGFR, transferrin, and low-density lipoprotein receptors [39] as well as membrane proteins such as the SARS-CoV-2 receptor ACE2. This pathway involves the assembly of the clathrin polyhedral lattice beneath the membrane and the association of tetrameric adaptor proteins, which bind to the sorting signals expressed on the cytoplasmic side of the virus receptor proteins. Many viruses have been shown to use the clathrin pathway, although different mechanisms may be used by a given virus in different cells, and multiple mechanisms may be exploited within a given cell type [39,40]. The dependency of SARS-CoV-2 on TMPRSS2 cleavage of the spike protein has suggested that fusion of the virus membrane with the plasma membrane was a major route of infection, bypassing the requirement for endocytosis. However, most coronaviruses are endocytosed prior to infection and SARS-CoV-2 has now been shown to be internalised through CME after binding to ACE2 [41,42]. The ACE2 receptor expresses a PDZ binding motif on the cytoplasmic side of the plasma membrane which serves as an endocytic motif for the clathrin adaptor subunit AP2. The AP2 adaptor is required for optimal formation of the clathrin lattice, although there are a number of other clathrin adaptors (e.g., NUMB and EPS15) which can mediate CME depending on the internalisation signals expressed by different membrane proteins. ACE2 is not the sole SARS-CoV-2 receptor, others such as some integrins [43], neuropilin 1 [44], and mGluR2 [45] have been implicated as receptors for SARS-CoV-2 and also contain PDZ binding domains.

Perhaps the best-characterised regulators of CME are the Numb-Associated Kinases (NAKs), AP2-associated protein kinase 1 (AAK1) and cyclin G-associated kinase (GAK). These kinases stimulate cargo recruitment [46], as well as the assembly and internalisation of the clathrin/AP2/cargo complex [47]. AAK1 and GAK are also involved in the further intracellular trafficking of the endocytic vesicle including removal of the clathrin coat [48,49,50]. As far as viruses are concerned, AAK1 and GAK may also regulate the infectivity of viruses that depend on AP2, such as the flaviviruses (e.g., Dengue and West Nile viruses), HCV, HIV, and Ebola virus [51,52,53,54].

## 3. The Role of AI in the Repurposing of Baricitinib

In January 2020, when it had become apparent that the new coronavirus was likely to spread worldwide, scientists at BenevolentAI, a London-based AI-enabled drug discovery company, used their AI-enhanced knowledge graph to piece together the mechanisms behind SARS-CoV-2 and then search for approved drugs capable of treating those mechanisms and thereby treat patients with the disease. This knowledge graph combines numerous data sources incorporating information on drugs, drug targets, genes, biological mechanisms, and diseases [55]. The knowledge graph contains machine-read literature covering the extensive collection of biomedical knowledge available (more than 30 M papers are catalogued in PubMed) and the contents of dozens of structured databases. The interrelationships between biomedical concepts in the knowledge graph are enhanced by AI algorithms, which help describe their confidence, causality, and cover gaps in established knowledge. The knowledge graph can be searched or explored by experts using interactive tools, as well as novel proprietary algorithms as described in [55]. To analyse SARS-CoV-2 mechanisms and identify candidate drugs, BenevolentAI scientists adopted an interactive and iterative approach, combining their expertise and interactive tooling with AI-generated biomedical relationships extracted from recent coronavirus literature. The aim was to find an approved drug which could treat the “cytokine storm” responsible for many of the early deaths from COVID-19 [56]. In addition, the drug should also prevent or reduce virus infection, perhaps by inhibiting the infection of cells by the virus which was thought to be via the SARS receptor ACE2.

The BenevolentAI knowledge graph is focused on human biology, so the search in January 2020 was focused on identifying drugs acting on host proteins that were subverted by the virus. In brief, the virus interactome was identified, added to the knowledge graph, and the knowledge graph was queried for anti-inflammatory agents which could counter the cytokine storm and also have antiviral effects. Since virus replication is largely mediated by proteins encoded by the virus, and the knowledge graph is focused on human biology, the search was for those mechanisms and proteins of the host mediating viral infection of cells rather than viral replication. In the search of this SARS-CoV-2 enhanced BenevolentAI knowledge graph for endocytic mechanisms a cluster of protein interactions related to virus entry suggested CME was the likely route of SARS-CoV-2 entry into cells (Figure 2) [56]. A Protein–Protein Interaction (PPI) output that indicated that SARS-CoV-2 may infect cells via CME is shown in Figure 2, where the CME module is identified. This figure illustrates how CME was identified as the probable entry pathway for the SARS-CoV-2 virus. This conclusion was later confirmed [41,42]. The proteins of the CME pathway (AAK1, CLTC, GAK, EPS15, AP2M1 etc.) were enriched in the pink endocytosis cluster of PPIs in Figure 2.

By focusing on anti-inflammatory and viral infection mechanisms as described in [56], the knowledge graph and computational tools revealed approved drugs potentially able to act as both anti-inflammatories and antivirals. The result of this process was the identification of two drugs, baricitinib (Figure 3) and fedratinib, approved for inflammatory indications, and ruxolitinib for myeloproliferative diseases. These were predicted inhibitors of JAKs and also of NAKs. Being JAK inhibitors, all three were likely to be effective inhibitors of cytokine signalling and complement activation and neutrophil trapping [35], thereby reducing the inflammatory consequences of the elevated levels of cytokines typically observed in people with COVID-19. Comparison of their pharmacokinetic properties, however, revealed that baricitinib was also predicted to inhibit the NAK enzymes AAK1 and BMP2K at plasma exposures routinely achieved when dosing patients. In contrast, the predicted unbound plasma exposures of ruxolitinib and fedratinib required to inhibit these enzymes (and so CME) greatly exceeded the exposures achieved therapeutically [57]. These drugs were, therefore, unlikely to reduce viral infectivity at tolerated doses, although they might reduce the host inflammatory response through JAK inhibition.

The combination of the oncology therapeutics sunitinib and erlotinib was previously shown to reduce the infectivity of a wide range of viruses, including Hepatitis C virus, Dengue virus, Ebola virus, and respiratory syncytial virus [52,58]. However, sunitinib and erlotinib would be difficult for patients to tolerate at the doses required to inhibit AAK1 and GAK so were not considered further. The high affinity of baricitinib for NAKs, its anti-inflammatory properties, its advantageous pharmacokinetic properties, and mild side effect profile (see later) suggested that it should be used in the treatment of COVID-19 [57,59].

## 4. Baricitinib in COVID-19 Therapy

After publishing the output of this AI-augmented research, the mechanistic predictions were validated, confirming that baricitinib inhibited signalling by a range of cytokines associated with the COVID-19 hyperinflammation. The anti-inflammatory effects of JAK inhibitors in general are summarised in ref. [35], and the effect of baricitinib confirmed in ref. [60] where the signalling of IL-2, IL-6, IL-10, IFNγ, and GCSF in monocytes, NK, and T cells was demonstrated [60]. Baricitinib also caused a significant reduction in plasma IL-6 in rheumatoid arthritis patients [60,61], these observations together indicating the potential of this drug to inhibit the hyperinflammation associated with COVID-19 (Figure 3). In addition, the nM potency of baricitinib on the NAK enzymes was confirmed in [60] and baricitinib was shown to reduce SARS-CoV-2 infection of human liver cells through super resolution microscopy, thereby confirming the predicted antiviral activity of this drug [62]. Perhaps as important was the observation that baricitinib reduced the expression of ISGs associated with platelet activation, suggesting it may reduce the extensive microthrombosis observed in COVID-19.

Baricitinib has other advantages including an oral once per day formulation, a predominantly renal route of clearance and low plasma protein binding. These properties suggested that baricitinib could be readily dosed with the antivirals being tested at the start of the pandemic, since they were largely cleared through liver metabolism. This enabled the testing of a combination with remdesivir in the ACTT-1 trial as well as with drugs being used as standard care.

### 4.1. Observational Clinical Trials

The first clinical test of baricitinib in COVID-19 was in four patients in Milan, all of whom recovered well [60]. Importantly, these patients underwent seroconversion while taking baricitinib, suggesting that this immunomodulator was unlikely to compromise the endogenous fight against infection. Almost simultaneously in Northern Italy, other hospitals were demonstrating that baricitinib appeared to reduce COVID-19 mortality. These and other small trials showed significant reductions in mortality and/or an improvement in lung function [63,64,65,66]. The next phase of validation was in propensity matched trials in Italy and Spain, which showed a substantial reduction in mortality associated with baricitinib treatment [62]. In 83 patients with moderate-severe SARS-CoV-2 pneumonia and including an aged cohort, baricitinib in the presence of the Standard of Care (SoC, which at the time included hydroxychloroquine, Kaletra, as well as glucocorticoids) caused a 71% reduction in mortality, with few drug-induced adverse events. Similarly, another propensity matched retrospective study showed a 48% reduction of mortality in patients over 70 years old [67].

These data helped to convince the National Institute for Allergy and Infectious Diseases of the NIH that baricitinib should be tested in a randomised trial. Since remdesivir had already shown a positive outcome in the ACTT-1 trial, (i.e., an increased rate of recovery with a 25% reduction of mortality by day 29 [30]), baricitinib was first tested in combination with remdesivir, comparing with the effect of remdesivir alone (ACTT-2). This was despite the fact that there has been some concern about the efficacy of remdesivir resulting in the WHO later not recommending its use in severe COVID-19 (a meta-analysis of four RCTs suggested no effect on mortality [68]).

### 4.2. Randomised Clinical Trials

The outcome of the ACTT-2 trial with over 1000 patients was a significant reduction in mortality and accelerated recovery in patients treated with baricitinib plus remdesivir compared with those treated with remdesivir alone. These effects were seen most strongly in those requiring supplemental oxygen (ordinal group 5) or non-invasive ventilation (ordinal group 6) at baseline. Both groups showed a 40% reduction in mortality [69]. Based on this and the aforementioned smaller non-randomised trials, the FDA issued an EUA authorising the use of this combination for the treatment of hospitalised COVID patients. Subsequently the CoV-BARRIER randomised trial reported the effect of baricitinib with standard of care (SoC) versus placebo with SoC [70]. Approximately 80% of the over 1500 patients in this trial were treated with dexamethasone, and 20% received remdesivir, as part of the SoC protocols in many institutions. Baricitinib had no effect on disease progression (primary end point) but, consistent with ACTT-2, reduced mortality by 38%. This life-saving effect was especially strong in those in ordinal scales 5 and 6 at baseline and was seen in those taking steroids or remdesivir or neither. Although it was considered possible that later stages of disease would be relatively resistant to such treatments, largely due to the amount of alveolar damage and hyaline membrane formation, in later analysis a 46% reduction in mortality was seen in patients receiving invasive mechanical ventilation (IMV) in this randomised trial [71]. It has been suggested [72] that remdesivir may have some synergistic effects with baricitinib, particularly on duration of disease, although the large life-saving effect of baricitinib is seen in the presence or absence of remdsivir [69,70]. Baricitinib is therefore the first immunomodulator to be shown to reduce COVID-19 mortality in a placebo-controlled trial [72]. Recently the RECOVERY trial examined the efficacy of baricitinib in comparing baricitinib plus SoC versus SoC in more than 8000 patients. In this trial, the mortality was significantly less than that seen in earlier RECOVERY trials, probably reflecting improved care and the widespread use of glucocorticoids (95%), remdesivir (20%), and tocilizumab (32%) in these patients. Despite this, baricitinib significantly reduced mortality (HR 0.87), with an approximate 25% reduction in mortality in those with severe disease (requiring ventilation at baseline) [73]. The effect on mortality was approximately half that seen in the previous trials, almost certainly because of the co-medications, although this interpretation awaits publication of the final data.

Studies of new therapies in at-risk individuals with early stage COVID-19 (prehospitalisation) have been difficult to execute because of the need for strict patient isolation, which precludes careful monitoring. Advanced contactless monitoring methods are being developed to enable such trials to be undertaken [28], including, for example, using combinations of effective oral antivirals such as nirmatrelvir/ritonavir (Paxlovid) with oral agents that limit inflammation-induced damage, such as baricitinib, in those whose risk is high due to disease or age. 

It is probable that this approach of targeting the host pathways subverted by the virus could be applied to other infections. For instance, flavivirus (e.g., Dengue) infections also show an early relatively mild disease which, in some cases, is followed by an aberrant hyperinflammation with vasculopathy. The Dengue viruses (and their NS1 proteins) access cells predominantly through CME [39], so anti-inflammatory agents which also inhibit CME (such as baricitinib) could be useful in these infections.

### 4.3. Baricitinib Safety

Long-term treatment (months to years) with JAK inhibitors is associated with significant side effects in a small number of patients [74]. These include an increased propensity for venous thromboembolisms and Herpes infections. However, in the clinical trials assessed for European Medicines Agency registration (which covered 4214 patient years of dosing), the most significant side-effect seen was a small increase in upper respiratory tract infections (similar to those observed with methotrexate). The incidence of serious infections (including Herpes zoster) over 52 weeks dosing was small (3.2 per 100 patient-years), and similar to placebo [75]. However, even these side effects were considered unlikely given the short duration of dosing required in COVID-19 treatments, and no such safety signals have since been observed in baricitinib COVID-19 trials. There were significantly reduced infections in the baricitinib arm in the ACTT-2 trial [69], while two other RCT reports showed no differences in adverse events between the baricitinib and control arms [70,76]. In the large RECOVERY trial there were no significant increases in thrombosis, secondary infections, or other safety outcomes in those treated with baricitinib and a significant reduction in mortality was observed [73]. Similarly, baricitinib was shown to cause significantly fewer adverse events in the ACTT-4 trial when compared with dexamethasone, both in the presence of remdesivir [77]. These small improvements in safety were also seen in a meta-analysis covering 3564 patients in a mixture of RCT and observational trials where serious adverse events and secondary infections were approximately 20% fewer in those treated with baricitinib [76]. It can therefore be concluded that baricitinib is a safe treatment for those hospitalized with COVID-19.

## 5. Comparison of Baricitinib and Other Immune Modulators

The WHO has a live document published in the BMJ in which its recommended therapies for COVID-19 are summarised [31]. In summary the WHO currently recommends that remdesivir, ivermectin, hydroxychloroquine, lopinavir–ritonavir, and convalescent plasma should not be used to treat severe or critical patients, whereas low dose corticosteroids, baricitinib, and tocilizumab should be used. Tofacitinib is also recommended in the absence of available baricitinib. This prevalence of recommended immune modulators is consistent with the biphasic nature of COVID-19 (Figure 1) and the concept that immune dysregulation is the major cause of mortality. Comparison of these immune modulators shows that baricitinib has the greatest effect on mortality (Table 1).

### 5.1. Other JAK Inhibitors

During this period, two other JAK inhibitors were tested in randomised COVID-19 trials, ruxolitinib and tofacitinib, neither of which was predicted to inhibit the NAK enzymes at therapeutic plasma concentrations. Early observational trials of the JAK1/3 inhibitor tofacitinib [81,82] and the STOPCOVID randomized trial showed reduced mortality, the latter with a HR of 0.63 in 289 patients [83]. Meta-analyses of observational and randomised trials with these JAK inhibitors indicated significant benefits on mortality, with HR values of 0.42 [84] and 0.12 [85]. These analyses included the JAK1/2 inhibitor ruxolitinib, (approved for the treatment of myeloproliferative diseases), which was effective in Phase 2 COVID-19 trials [86]. However, this was not confirmed in the larger, randomised Phase 3 trials (RUXCOVID and RUXCOVID-DEVENT) when no effect on mortality of hospitalised patients was seen. This is surprising given the efficacy of baricitinib and tofacitinib but could reflect the ability of baricitinib to inhibit the NAK enzymes (and so reduce virus infectivity). Interestingly, when considering severe and critically patients (as in the RUXCOVID-DEVENT trial and subgroup analyses of the COV-BARRIER and RECOVERY trials) [73], the JAK inhibitors have been remarkably effective (HR of 0.41, 0.47, and 0.74 respectively).

### 5.2. Low Dose Glucocorticoids

Low-dose dexamethasone has been widely used since the interim analysis of the RECOVERY clinical trial revealed its beneficial effects [87]. Later analysis of the whole RECOVERY study showed a HR of 0.83 for low-dose dexamethasone [87]. Subsequently, the WHO REACT team reported a HR to death or IMV of 0.7 for those treated with glucocorticoids [88], an analysis which included the data from the interim analysis of the RECOVERY trial covering approximately 1000 patients with IMV indicating a HR of 0.59. These data exemplify the common observation that glucocorticoids, and other immune modulators, have a greater effect on severe and critical disease consistent with the role of the aberrant immune system in mortality. One issue with the use of glucocorticoids is the observation in the RECOVERY trial that patients with mild disease (i.e., not requiring oxygen supplementation) showed increased mortality on treatment. There have also been some observational trials suggesting increased or no effect on mortality of low dose glucocorticoids, and it is now recommended that glucocorticoids should only be used in patients with severe or critical disease [31,89]. It is also noticeable that even in the presence of dexamethasone, baricitinib, tofacitinib, and tocilizumab have all been shown to confer enhanced protection in randomised SoC-controlled clinical trials. There have not been many direct comparisons of the immune modulators in COVID-19 trials, except in the recently published ACTT-4 trial where the effect of baricitinib with remdesivir plus SoC was compared with dexamethasone plus remdesivir and SoC [77]. In this study there was no difference in survival of patients requiring supplemental oxygen, but there were significantly more grade 3 or 4 adverse events in the dexamethasone group.

### 5.3. IL-6 Receptor Blocking Antibodies

During the hyperinflammation phase of COVID-19, IL-6 is frequently the most highly elevated cytokine in the plasma. This led to the testing of the anti-IL6 receptor monoclonal antibody tocilizumab in multiple trials in 2020, leading to its approval for the treatment of COVID-19 in the US, UK, and EU, and being recommended by the WHO. Significant improvements in mortality were seen in the REMAP-CAP [90] and RECOVERY [91] trials. In contrast, no effect on mortality was seen in multiple smaller trials including COVACTA [92], EMPACTA [93], or BACC Bay [94] and dosing with remdesivir [95]. Interestingly, despite no effect on mortality in the CORIMUNO-TOCI-1 trial [96], later analysis showed that those with high plasma CRP did benefit from the treatment with tocilizumab [97]. This is consistent with those experiencing hyperinflammation (as reflected by high CRP) benefitting from IL-6 receptor inhibition. In contrast, there was no effect on mortality in a 420 patient RCT of sarilumab, another anti-IL-6R antibody [98]. The difference in outcome with these two agents has yet to be explained. As in the clinical trials testing baricitinib and tofacitinib, most of the patients in these tests of the anti-IL-6R agents were also receiving glucocorticoids.

### 5.4. IL-1 Blocking Antibodies

IL-1β is another cytokine frequently elevated in COVID-19 patients, and the anti-IL-1 receptor antibody anakinra has been tested in clinical trials. In a meta-analysis covering 895 patients in mostly observational studies, a significant effect on mortality was seen when dosed in the absence of steroids (HR 0.32). Surprisingly, this analysis suggested anakinra was of no further benefit when given with steroids [99]. In a highly selected patient population, an RCT involving 594 patients with elevated soluble urokinase plasminogen receptor (suPAR) at baseline clearly showed a significant reduction in mortality (HR 0.45) with anakinra [100]. Soluble uPAR is a biomarker of COVID-19 disease progression, including respiratory failure [101] and kidney injury [102]. It is also a component of the senescence-associated secretory phenotype [7] and a marker of systemic inflammation, including that associated with aging [103]. A similar approach using canakinumab, an anti-IL-1β antibody, in patients with elevated CRP did not, however, show a significant effect on mortality, although a trend was apparent [104]. There are a number of possible explanations for these observations including a possible requirement for IL-1α inhibition (as seen with anakinra) and the use of different biomarkers in these trials to identify likely susceptible populations [32].

Despite the variability in the outcomes of the studies in Table 1, and the difficulty in comparing efficacy between clinical trials, it is clear that these immune modulators are effective treatments for hospitalised patients with COVID-19. In addition, it appears that the JAK inhibitor baricitinib is the most effective treatment, even conferring added benefit in the presence of antivirals and other immune modulators. The ACTT-4 trial, which showed no difference between baricitinib and dexamethasone in the presence of remdesivir, is consistent with one advantage of baricitinib, namely, its anti-viral properties [77].

## 6. Conclusions

In summary, baricitinib has shown the greatest mortality benefit in large numbers of COVID-19 patients in randomised clinical trials [72] while tocilizumab and low-dose corticosteroids have shown smaller but significant effects. There are caveats to these data, however. The lack of efficacy of the JAK1/2 inhibitor ruxolitinib in Phase 3 trials is surprising, although a post hoc analysis of the DEVENT trial suggested a reduction in mortality in a subgroup of patients. Similarly, the failure of the anti-IL-6 receptor antibody sarilumab has yet to be explained, while the efficacy of the low-dose glucocorticoids is heavily dependent on data from a single trial (RECOVERY). In addition, while glucocorticoids should be used sparingly in patients with mild disease, baricitinib and the other cytokine modulators had no deleterious effects in such patients [69,70]. It should be noted that baricitinib, tocilizumab, and tofacitinib all showed reduced mortality even in the presence of low-dose steroids, indicating that glucocorticoids are insufficient when used without at least one of these more focused modulators of cytokine action. The side effect profiles of these agents are mild, permitting their widespread use in multiple populations. Baricitinib, tofacitinib, and the glucocorticoids are relatively inexpensive when given for the typical two weeks and are easy to dose, transport, and store, in contrast to the antibodies such as tocilizumab. This makes them good candidates for use in low- and middle-income countries as well as candidates for use in early intervention trials. For instance, baricitinib could be tested in at-risk patients, provided patients are accurately monitored for safety using contactless digital methods. 

From all of this, it is clear that, as recommended by the WHO, baricitinib or tocilizumab with low-dose glucocorticoids are the most effective therapies currently available for severely ill patients. It is also probable that the success seen with the combined antiviral and anti-inflammatory effects of baricitinib could lead to the effective treatment of other viral diseases (e.g., DENV, WNV etc.) characterised by clathrin-mediated infection of target cells and hyperinflammation.

In conclusion, baricitinib has proven to be the best available drug for COVID-19, limiting the inflammatory phase of the disease and reducing the risk of progression to advanced disease. It has been approved by the World Health Organization and has entered standard of care guidelines in many centres. It represents an example of the power of AI to identify useful drugs and is a model for the rapid discovery of potentially useful drugs in future pandemics. Finally, baricitinib is an example of a drug that limits disease independent of host immunity, and such drugs will be essential if new, highly pathogenic SARS-CoV-2 variants evade existing immunity or if similar viral pandemics occur in the future.

## Figures and Tables

**Figure 1 vaccines-10-00951-f001:**
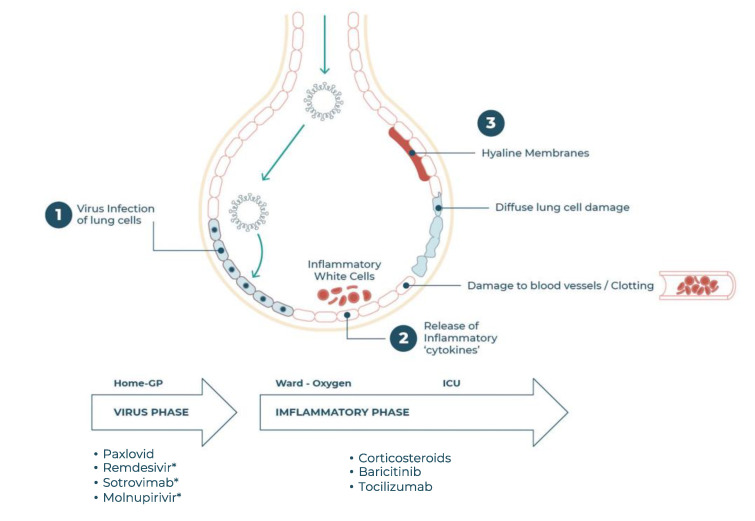
Illustration of the biphasic nature of COVID-19. Infection and replication of virus in lung epithelial cells being the first phase, followed by innate immune cell recruitment, inflammation, and resultant tissue damage being the second phase. The drugs strongly recommended by the WHO for the treatment of the Omicron COVID-19 variants in the different phases are indicated, while * indicates a conditional recommendation for those at high risk of severe disease.

**Figure 2 vaccines-10-00951-f002:**
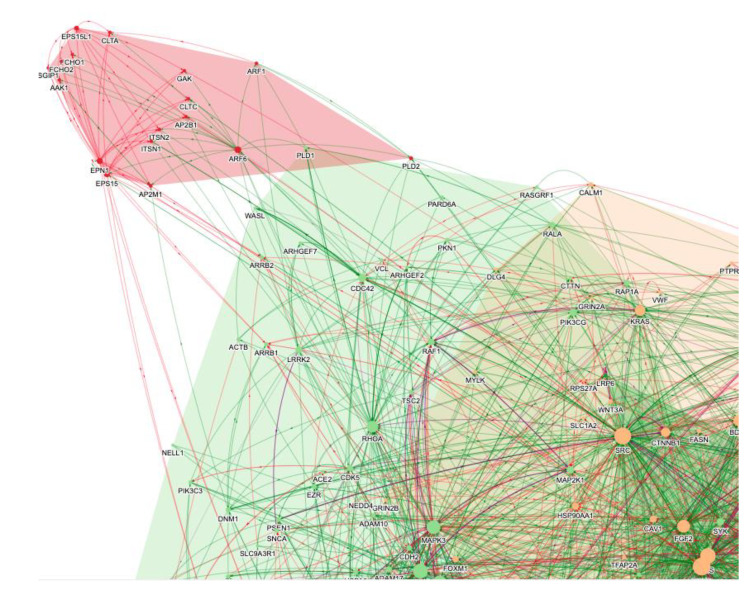
A selection of the PPI networks from a knowledge graph query of “SARS-CoV-2 AND endocytosis”. Specific pathways and processes are grouped in different-coloured clusters (e.g., endocytosis in pink and cytokine signalling in green and orange). Each node reflects one protein, and the edges reflect enhancing (green) or inhibiting (red) protein–protein interactions. The CME module is in pink.

**Figure 3 vaccines-10-00951-f003:**
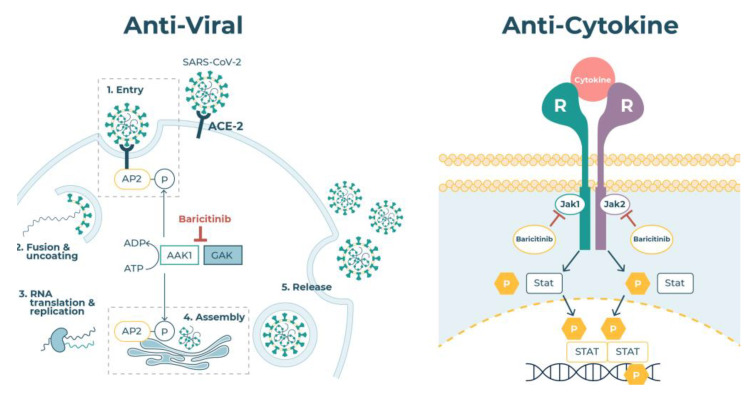
Baricitinib has both antiviral and anti-inflammatory properties. The BenevolentAI knowledge graph was used to identify CME as the probable route of SARS-CoV-2 infection, and baricitinib as a potential inhibitor of this process through the inhibition of the NAK enzymes, particularly AAK1. The well-known anti-inflammatory effect of baricitinib complements this through the inhibition of cytokine action.

**Table 1 vaccines-10-00951-t001:** Meta-analyses of immune modulators in randomized and controlled COVID-19 clinical trials.

	Mortality (%) SoC Treated		HR	Patients (n)	Studies	Reference
Baricitinib ^1^	13.6	7.3	0.56	3827	9	[76]
Baricitinib ^2^	13.3	11.3	0.69	10,815	4	[78]
JAK inhibitors ^3^	14.5	11.7	0.80	11,888	9	[73]
Glucocorticoids	31.1	27.3	0.85	6250	7	[79]
Tocilizumab	25.8	21.8	0.86	6311	8	[80]
Sarilumab	18.5	21.1	1.08	2826	9	[80]

^1^ The baricitinib data meta-analyses included two Phase 3 studies (2558 patients) [70,71] also included in the JAK inhibitor meta-analysis. ^2^ This meta-analysis included RCTs only. ^3^ The JAK inhibitor meta-analysis included baricitinib, tofacitinib, and ruxolitinib trials as well as the baricitinib Phase 3 studies.

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
