# Peer review of "The AI-Assisted Identification and Clinical Efficacy of Baricitinib in the Treatment of COVID-19"

_vaccines, 2022, doi:10.3390/vaccines10060951_

Round 1

Reviewer 1 Report

Some COVID-19 patients  had a potentially fatal aberrant hyperinflammatory immune reaction characterized by high levels of IL-6 and other cytokines. Baricitinib is a Janus Kinase (JAK) 1/2 inhibitor approved  for rheumatoid arthritis. In this  review the AI assisted identification of baricitinib, its antiviral and anti-inflammatory properties  and efficacy in clinical trials is discussed and compared with that of other immune modulators including glucocorticoids, IL-6 and IL-1 receptor blockers and other Janus Kinase inhibitors. Baricitinib inhibits both virus infection and cytokine signalling, in both observational and randomized clinical trials, both alone and in combination with glucocorticoids and IL-6 receptor antibodies significantly reduced mortality, had few side effects, and were non-immunological.

Will be essential if new, highly pathogenic SARS-CoV-2  variants evade existing immunity or if similar viral pandemics occur in the future.

Author Response

We thank reviewer for a very accurate summary of the paper in which we indeed aimed to show how AI was used to identify baricitinib, illustrate the mechanisms by which it may operate and demonstrate that baricitinib was effective in the treatment of hospitalised COVD-19 patients. 

Reviewer 2 Report

This paper talks about using baricitinib in treating COVID-19. The paper describes the relationship between host and the virus first from the perspective of aging and immune responses, and then delineates the mechanisms of JAK inhibitors as well as virus endocytosis which are related to baricitinib. The paper then covers baricitinib from artificial intelligence’s aspect and shows clinical data of baricitinib and related drugs in COVID-19 treatments. This paper contains information related to the therapy of COVID-19 which has significant meanings in this specific pandemic time. Please check the comments below for improvements:

  1. This paper is more than “baricitinib in the treatment of COVID-19”. Please reconsider your title.
  2. Please double check all your citations. For example, 165-174, 187-194, 217-229, 231-238, 361-366, 435-438, 456-462, lack citations. Usually, we want to see citations after each of the solid statements that describes facts or results that are mentioned previously in others’ studies.
  3. Figure 2 doesn’t make much sense here. It’s important to describe at least what the colors of the lines and spaces are in order to emphasize why the zoomed in part is highlighted. The whole figure, together with the zoomed in part, contains too much un-explained information.
  4. Please refer the figures in your texts so when readers read your paper, they know why the figures are there and what the figures are explaining.
  5. English needs proofreading. Like in line 280, it’s ‘COVID-19’ (dash missing) and ‘therapy’ (not theapy). Please go over your manuscript carefully and make changes accordingly.
  6. When you talk about using Baricitinib in COVID-19 therapy, mention the details on mechanisms again by explaining more on your figure 3 legends.
  7. Please talk more on the safety of baricitinib since drug safety is so important that no matter how good the drug is, if there’s strong side effects, the drug cannot be used. This part could be expanded by focusing not only on baricitinib; drugs related to it can also be used for safety discussion.

Author Response

The reviewer's points are well made and we have adapted the text to take them into account

  1. A new title: The AI assisted identification and clinical efficacy of baricitinib in the treatment of COVID-19
  2. Citations have now been included to support the statements on lines 165-174 (new citation on line 175), 187-194 (line 193), 217-229 (line 227), 231-238 (lines 234 and 252), 361-366 (line 382).  For the statements in lines 435-438 the relevant citations are in the previous 10 lines and for 456-462 the relevant references are cited in Table 1.
  3. The Figure2 legend has been improved to describe the clusters, the nodes and edges and reference to why the Figure is included in lines 252-257. This Figure with lines 238-249 shows in a simple summarised way how CME was identified as the probable SARS-CoV-2 infection route in human cells and we think this important for our readers to understand in outline the process of identification of this drug. For those who are interested references 54 and 55 have the details.
  4. New referral to Figure 3 in line 268, to Figure 2 in line 253 and Figure 1 in lines 43 and 413 
  5. Therapy and COVID-19 have been corrected, the JAK and AI abbreviations regularised and numerous improvements (as shown in track changes) to the English made
  6. The legend in Figure 3 has been expanded as suggested
  7. The safety aspects have been expanded in section 4.3.  showing that baricitinib is safe in COVID-19.

Other modifications: the ACTT-4 publication is now referenced in both safety and comparison with dexamethasone

Reviewer 3 Report

The title of the manuscript  suggests that   this review will address the role of baricitinib in COVID-19. Instead, an expanded analysis of the immune response of the host is provided  followed by a very epidermic look in other agents against COVID-19. The rejection of remdesivir and the omittment from  figure 1 is far from being true ( see also  Lancet 2022; 3-day remdesivir N Engl J Med 2021 and curent EMA, IDSA treatmen guidelines). The allocation of  a potential the synergistic effect of baricitinb with remdesivir in  the ACTT-2 trial  was already mentioned in another review by the same  last author (Lancet Respir Med 2021), but not in the current one. The very analytical impact of artificial intelligence in the eligibility of JAK inhibitors for the treatment of COVID-19  is disiorienting for the reader. Instead, a figure pointing out all the potential ways of action of the drug would be more than useful ( eg. Stebbing J EMBO Mol Med (2020)12:e12697).  In conclusion, this paper should be revised

Author Response

We are grateful to this reviewer since the article obviously needed correcting to increase understanding. We have changed the title and made clearer the aims of the article in the first paragraph of the Introduction.  

We apologise for omitting remdesivir from Figure 1, this has now been rectified with a modified Figure and legend, updated with the latest WHO recommendations.  The comparison of steroids +remedesivir and baricitinib + remdesivir (ACTT4) has also now been included following its recent publication (line 400-405 and 452 and 495)

The possible synergism between remdesivir and baricitinib is now referred to on line 354.

We think that the contribution of AI was central to the identification of baricitinib for COVID19 and so believe it should remain in the article. 

We have now increased the description of the anti-inflammatory effects, and they are now referred to references 34 and 59 for details (lines 291-300).

Round 2

Reviewer 3 Report

The current form of the manuscript is a  well-comprehensive and complete review of  the rationale of baricitinib administration in patiens with COVID-19.

 No additional comments.